# Design, Synthesis and Biological Evaluation of Lophanic Acid Derivatives as Antifungal and Antibacterial Agents

**DOI:** 10.3390/molecules27206836

**Published:** 2022-10-12

**Authors:** Xiang Yu, Xun Song, Yi Zhang, Yemeng Yang, Jianghai Ye, Yahua Liu, Lutai Pan, Hongjie Zhang

**Affiliations:** 1The Key Laboratory of Miao Medicine of Guizhou Province, School of Pharmacy, Guizhou University of Traditional Chinese Medicine, Guiyang 550025, China; 2School of Chinese Medicine, Hong Kong Baptist University, Hong Kong 999077, China; 3College of Pharmacy, Shenzhen Technology University, Shenzhen 518118, China

**Keywords:** lophanic acid, antifungal activity, antibacterial activity, structural modification

## Abstract

In order to discover more promising antifungal and antibacterial agents, a series of new derivatives were designed and synthesized by structure modification based on the naturally occurring antimicrobial compound lophanic acid. The structures of all the target compounds were well characterized by spectroscopic data. The stereochemistry of these compounds was further determined through the X-ray diffraction analysis of 6a. The synthetic compounds were evaluated for their antimicrobial activities against filamentous fungi (*T. rubrum, T. mentagrophytes*), yeasts (*C. neoformans, C. albicans*) and Gram-positive and Gram-negative bacteria (*MRSA, S. mutans, S. sobrinus,* and *E. coli*). Among them, 3d and 3i are found as the most promising leads that showed potent inhibitory effects against all the tested fungal and bacterial strains except for *E. coli*. The presence of the C-20 carboxylic ester groups and the free hydroxy group at C-13 was found to be essential for the antifungal and antibacterial activities of the lophanic acid derivatives.

## 1. Introduction

Tinea pedis is a chronic or recurring disease characterized by dermatophytic infection of the feet and toes, which can involve the interdigital web spaces or the sides of the feet, and it is often caused by anthropophiles, including *Trichophyton rubrum* sensu stricto, *T. interdigitale* and *Epidermophyton floccosum* [1]. Currently, the effective treatments for tinea pedis are topical or oral antifungals or a combination of both, such as terbinafine and clotrimazole [2,3]. Although synthetic chemical drugs have been used to treat tinea pedis, their overuse over the years has led to considerable concern for fungus resistance and other adverse effects on human health [4,5]. Thus, there is an urgent need to discover new promising alternatives from sustainable natural bioresources to effectively and selectively treat tinea pedis.

For decades, medicinal plants have been considered a rich source of lead compounds for drug discovery and development [6,7]. Abietane-type diterpenoid is one of the most prevalent classes of diterpenes widely found in the plants of the *Isodon* genus [8,9]. They have shown a variety of pharmacological activities, including anticancer, anti-inflammatory, and antivirus activities [10,11]. Meanwhile, abietane-type diterpenoids were also found to show exceeding antimicrobial activities. For example, our previous studies described that rubesanolide D (Figure 1I) had antibacterial activity against biofilm formation of the dental bacterium *Streptococcus mutans* [12], kunminolide A (Figure 1II) and fladin A (Figure 1III) displayed inhibitory effects against the dental pathogens *S. mutans*, *Porphyromonas gingivalis* and *Candida albicans* and the athlete’s foot fungus *T. rubrum* [13,14]. Abietane-type diterpenoids appear to be attractive molecules for further structure modification to discover novel antifungal and antibacterial agents.

Lophanic acid (**1**, Figure 1), a naturally occurring abietane-type diterpenoid, was found abundant in the medicinal plants *I. flavidus* [13] and *I. lophanthoides* [14]. Our study has demonstrated that the compound possesses potent biological activities against a wide range of fungi and bacteria, including *T. rubrum, P. gingivalis, S. mutans*, and *S. albus* [15]. However, little attention has been paid to the structural modification to synthesize the derivatives of **1** as antifungal and antibacterial agents. In our previous research on the bioactive compounds from *Isodon* plants, we obtained a rich amount of lophanic acid (**1**), which allowed us to carry out a synthetic study by using lophanic acid as a scaffold. Herein we report the design and synthesis of a series of new lophanic acid derivatives, and the antifungal and antibacterial activity evaluation of the synthetic derivatives against two filamentous fungi (*T. rubrum, T. mentagrophytes*), two yeasts (*C. neoformans, C. albicans*), and four Gram-positive and Gram-negative bacteria (*MRSA, S. mutans, S. sobrinus,* and *E. coli*).

## 2. Results and Discussion

### 2.1. Chemistry

As shown in Figure 1, fourteen C-20 ester derivatives (**3a**–**3n**) were first synthesized by reaction of lophanic acid (**1**) with the corresponding alkyl halides in the presence of K_2_CO_3_ at room temperature. The carboxyl group of **3a** was further reduced to an alcohol group by LiAlH_4_ to obtain compound **4**, which was substituted with different alkoxy groups at C-20 to produce **5a**–**5h**. In addition, as outlined in Figure 2, the hydroxyl groups of compounds **3a** and **3d** underwent an esterification reaction with acetyl chloride in the presence of pyridine to afford **6a** and **6b**. Interestingly, the acetyl group at C-13 of **6a** was further acetylated to form acetylacetic ester (**6c**) through a C–C bond. The structure determination of **6c** was evidenced by the compassion among the ^1^H NMR spectra of **3a**, **6a,** and **6c** as depicted in Figure 2. The stereochemistry of 6a was further confirmed by the X-ray crystallographic analysis (Figure 3).

### 2.2. In Vitro Antifungal Activity Evaluation against Filamentous Fungi and Yeasts

The antifungal activities of lophanic acid derivatives (**3a**–**6c**) were investigated against two filamentous fungi (*T. rubrum*, *T. mentagrophytes*) and two yeasts (*C. neoformans, C. albicans*) in vitro at 100 μg/mL with miconazole served as a positive control agent.

As shown in Table 1, compounds **3d** and **3i** exhibited a broad spectrum of antifungal activities against *T. rubrum, T. mentagrophytes,* and *C. neoformans* with inhibitory rates over 60%. On the other hand, some compounds were found to exhibit selective activities against filamentous fungi. For example, compound **3a** inhibited the growth of *T. rubrum* by 81.61%, but it showed only a 49.1% inhibitory rate against *C. neoformans*. Compounds **3b**, **3e**, **5c,** and **5h** displayed mild antifungal effects against *T. rubrum* and *C. neoformans*. By analyzing the structure-activity relationship (SAR) of the lophanic acid derivatives, we observed that the C-13 hydroxy is essential for retaining their antifungal activities. For example, the antimicrobial inhibitory rates of compounds **3a** and **3b** were measured over 60%, whereas their corresponding derivatives **6a**–**6c** showed almost no antimicrobial activities. Based on the antimicrobial activities of the lophanic acid derivatives (e.g., **3a, 3d**, **5a,** and **5b**), the structural modification at C-20 could be performed to improve the bioactivity. On the other hand, when a phenyl ring was introduced with an electron-donating group, the corresponding compounds exhibited better activity potency than those with an electron-withdrawing group. For example, **3i** showed growth inhibitory rates against *T. rubrum*, *T. mentagrophytes*, and *C. neoformans* at 74.91, 63.00, and 88.54%, respectively, while **3n** displayed no antimicrobial activities against these fungi. Interestingly, no lophanic acid derivatives were found to show antifungal activities against *C. albicans* at the concentration of 100 µg/mL.

### 2.3. In Vitro Antibacterial Activity Evaluation against Gram-Positive and Gram-Negative Bacteria

The antibacterial activities of compounds **3a**–**6c** were further tested against Gram-positive bacteria (*MRSA, S. mutans*, and *S. sobrinus*) and Gram-negative bacterium (*E. coli*) in vitro at the concentration of 100 μg/mL with tetracycline as the positive control agent (Table 2). Compounds **3a**, **3b**, **3d**, **3e**, **3i**, **5c**, and **5h** displayed more potent antibacterial activities against MRSA than they are against the other bacteria. Especially, **3b**, **3d**, **5c,** and **5h** were found to possess antibacterial activities with inhibitory rates greater than 90%. However, no antimicrobial activities were observed for the synthetic derivatives against *S. mutans, S. sobrinus*, and *E. coli*. Preliminary SAR analysis showed that the C-13 hydroxy plays an indispensable role in the antibacterial activities of the lophanic acid derivatives. For instance, compounds **3a** and **3b** showed strong activities with inhibitory rates of 84.43% and 95.89%, respectively. On the contrary, the introduction of an acyl group on the C-13 hydroxy group of **3a** and **3b** resulted in the loss of the antibacterial activities (e.g., **6a**-**6c**). Furthermore, a proper ester or alkyloxy group substituted at C-20 was found important for retaining the antibacterial activity potency of a lophanic acid derivative (e.g., **3a**, **3b**, **3d**, **5c,** and **5h**).

## 3. Materials and Methods

### 3.1. Chemistry

All chemical reagents were purchased and utilized without further purification. Solvents were used directly or treated with standard methods before use. Melting points (m.p.) were determined on an X-6a digital melting point apparatus (Gongyi Tech Instrument Co., Ltd., Gongyi, China) and were uncorrected. Infrared spectra (IR) were recorded on a Bruker TENSOR 27 spectrometer. Proton nuclear magnetic resonance spectra (^1^H NMR) and carbon nuclear magnetic resonance spectra (^13^C NMR) were recorded in CDCl_3_ on a Bruker Avance 400, 500, or 600 MHz instruments using tetramethylsilane (TMS) as the internal standard. High-resolution mass spectra (HRMS) were carried out with IonSpec 4.7 Tesla FTMS instrument. The purities of all the title compounds were determined on an UltiMate 3000 (Dionex, Sunnyvale, CA, USA) HPLC system and were of > 95% purity.

#### 3.1.1. General Procedure for the Synthesis of Compound **3a**–**n**

Lophanic acid (**1**, 100 mg, 0.31 mmol) and potassium carbonate (86 mg, 0.62 mmol) were dissolved in *N*, *N*-Dimethylformamide (DMF, 5 mL), and the solution was stirred at room temperature. Then a solution of substituent haloalkane (0.62 mmol) in DMF (2 mL) was added dropwise for 10 min. When the reaction was complete, checked by thin-layer chromatography (TLC) analysis, pure water (30 mL) was added to the reaction, which was extracted with ethyl acetate (3 × 30 mL). The combined organic phase was dried over anhydrous Na_2_SO_4_, filtered, concentrated under reduced pressure, and purified by silica gel column chromatography eluting with petroleum ether/ethyl acetate to afford compound **3a-n** in 85–98%.

Data for **3a**: White solid, yield: 98%, m.p. 99–101 °C; IR cm^−1^: 3527, 1709, 1213, 1141; ^1^H NMR (400 MHz, CDCl_3_): *δ* 3.61 (s, 3H, OCH_3_), 2.57 (d, *J* = 12.6 Hz, 1H), 2.27–2.13 (m, 3H), 2.07–1.98 (m, 2H), 1.89–1.81 (m, 3H), 1.72–1.65 (m, 3H), 1.59–1.51 (m, 3H), 1.42–1.36 (m, 2H), 1.26–1.20 (m, 1H), 0.98–0.94 (m, 1H), 0.92–0.91 (m, 6H, H-16, 17), 0.89 (s, 3H, H-19), 0.68 (s, 3H, H-18); ^13^C NMR (100 MHz, CDCl_3_): *δ* 176.2 (C-20), 130.1 (C-9), 129.6 (C-8), 72.0 (C-13), 52.4 (OCH_3_), 51.4 (C-5), 48.7 (C-10), 41.8 (C-3), 41.2 (C-14), 34.5 (C-1), 34.4 (C-4), 33.7 (C-15), 32.2 (C-12), 32.0 (C-7), 31.6 (C-18), 21.5 (C-11), 20.1 (C-2), 19.8 (C-19), 18.1 (C-6), 16.7 (C-16), 16.7 (C-17); HRMS *m/z* calcd for C_21_H_34_O_3_Na ([M + Na]^+^) 357.2400, found 357.2401 (Appendix A).

Data for **3b**: White solid, yield: 95%, m.p. 52–54 °C; IR cm^−1^: 3540, 1709, 1214, 1145; ^1^H NMR (600 MHz, CDCl_3_): *δ* 4.13–4.03 (m, 2H, CH_2_CH_3_), 2.56 (d, *J* = 12.0 Hz, 1H), 2.29–2.05 (m, 4H), 2.02–1.97 (m, 1H), 1.92–1.81 (m, 3H), 1.71–1.64 (m, 3H), 1.58–1.51 (m, 3H), 1.41–1.36 (m, 2H), 1.25–1.18 (m, 4H), 0.96–0.93 (m, 1H), 0.92–0.89 (m, 6H, H-16, 17), 0.90 (s, 3H, H-19), 0.72 (s, 3H, H-18); ^13^C NMR (150 MHz, CDCl_3_): *δ* 175.6 (C-20), 130.3 (C-9), 129.5 (C-8), 72.0 (C-13), 60.2 (CH_2_CH_3_), 52.4 (C-5), 48.7 (C-10), 41.9 (C-3), 41.2 (C-14), 34.5 (C-1), 34.5 (C-4), 33.8 (C-15), 32.2 (C-12), 32.0 (C-7), 31.6 (C-18), 21.5 (C-11), 20.1 (C-2), 20.0 (C-19), 18.2 (C-6), 16.8 (C-16), 16.8 (C-17), 14.2 (CH_2_CH_3_); HRMS *m/z* calcd for C_22_H_36_O_3_Na ([M + Na]^+^) 371.2557, found 371.2558 (Appendix A).

Data for **3c**: colorless oil, yield: 89%; IR cm^−1^: 3660, 1735, 1726, 1381, 1130; ^1^H NMR (600 MHz, CDCl_3_): *δ* 4.74 (d, *J* = 15.6 Hz, 1H, CH_2_COOCH_2_CH_3_), 4.32 (d, *J* = 15.6 Hz, 1H, CH_2_COOCH_2_CH_3_), 4.25–4.16 (m, 2H, CH_2_COOCH_2_CH_3_), 2.63 (d, *J* = 12.1 Hz, 1H), 2.26–2.12 (m, 4H), 2.06–2.00 (m, 2H), 1.85–1.78 (m, 2H), 1.76–1.64 (m, 3H), 1.56–1.51 (m, 3H), 1.47–1.39 (m, 2H), 1.27–1.20 (m, 4H), 1.04–0.99 (m, 1H), 0.96 (d, *J* = 6.6 Hz, 3H, H-16), 0.93 (d, *J* = 6.6 Hz, 3H, H-17), 0.90 (s, 3H, H-19), 0.71 (s, 3H, H-18); ^13^C NMR (150 MHz, CDCl_3_): *δ* 175.1 (C-20), 168.2 (CH_2_COOCH_2_CH_3_), 130.6 (C-9), 129.2 (C-8), 71.4 (C-13), 61.5 (CH_2_COOCH_2_CH_3_), 60.2 (CH_2_COOCH_2_CH_3_), 51.9 (C-5), 48.5 (C-10), 41.9 (C-3), 40.3 (C-14), 36.1 (C-1), 34.7 (C-4), 33.9 (C-15), 32.0 (C-12), 31.9 (C-7), 31.4 (C-18), 20.5 (C-11), 20.0 (C-2), 19.9 (C-19), 18.2 (C-6), 17.1 (C-16), 16.9 (C-17), 14.1 (CH_2_COOCH_2_CH_3_); HRMS *m/z* calcd for C_24_H_38_O_5_Na ([M + Na]^+^) 429.2611, found 429.2613 (Appendix A).

Data for **3d**: colorless oil, yield: 92%; IR cm^−1^: 1732, 1457, 1115; ^1^H NMR (600 MHz, CDCl_3_): *δ* 4.68 (s, 2H, CH_2_CN), 2.59 (d, *J* = 13.8 Hz, 1H), 2.21–2.13 (m, 3H), 2.09–2.01 (m, 2H), 1.87–1.74 (m, 4H), 1.67–1.56 (m, 5H), 1.45–1.42 (m, 2H), 1.25–1.19 (m, 1H), 1.03–0.98 (m, 1H), 0.89–0.91 (m, 9H, H-16, 17, 19), 0.71 (s, 3H, H-18); ^13^C NMR (150 MHz, CDCl_3_): *δ* 174.3 (C-20), 131.3 (C-9), 128.9 (C-8), 114.5 (CH_2_CN), 71.9 (C-13), 52.4 (C-5), 49.1 (CH_2_CN), 48.2 (C-10), 41.7 (C-3), 41.3 (C-14), 34.6 (C-1), 34.5 (C-4), 33.9 (C-15), 31.9 (C-12), 31.9 (C-7), 29.8 (C-18), 21.6 (C-11), 20.2 (C-2), 20.1 (C-19), 18.2 (C-6), 16.8 (C-16), 16.8 (C-17); HRMS *m/z* calcd for C_22_H_33_O_3_NNa ([M + Na]^+^) 382.2352, found 382.2352 (Appendix A).

Data for **3e**: colorless oil, yield: 97%; IR cm^−1^: 1716, 1456, 1205, 1129; ^1^H NMR (600 MHz, CDCl_3_): *δ* 7.26–7.35 (m, 5H, H-Ph), 5.16 (d, *J* = 12.6 Hz, 1H, PhCH_2_), 5.00 (d, *J* = 12.6 Hz, 1H, PhCH_2_), 2.58 (d, *J* = 13.2 Hz, 1H), 2.30–2.17 (m, 2H), 2.13–1.97 (m, 3H), 1.90–1.78 (m, 3H), 1.70–1.63 (m, 2H), 1.59–1.45 (m, 3H), 1.40–1.37 (m, 3H), 1.22–1.17 (m, 1H), 0.97–0.92 (m, 1H), 0.90 (s, 3H, H-16), 0.89 (s, 3H, H-17), 0.88 (s, 3H, H-19), 0.67 (s, 3H, H-18); ^13^C NMR (150 MHz, CDCl_3_): *δ* 175.5 (C-20), 136.3 (C-Ph), 130.1 (C-9), 130.0 (C-Ph), 128.6 (C-8), 128.1 (C-Ph), 128.0 (C-Ph), 71.9 (C-13), 66.2 (PhCH_2_), 52.5 (C-5), 48.9 (C-10), 42.0 (C-3), 41.5 (C-14), 34.6 (C-1), 34.0 (C-4), 33.9 (C-15), 32.1 (C-12), 32.0 (C-7), 32.0 (C-18), 21.8 (C-11), 20.2 (C-2), 20.2 (C-19), 18.3 (C-6), 16.8 (C-16), 16.8 (C-17); HRMS *m/z* calcd for C_27_H_38_O_3_Na ([M + Na]^+^) 433.2713, found 433.2713 (Appendix A).

Data for **3f**: colorless oil, yield: 85%; IR cm^−1^: 1715, 1513, 1224; ^1^H NMR (600 MHz, CDCl_3_): *δ* 7.30–7.26 (m, 2H, H-Ph), 7.03–7.01 (m, 2H, H-Ph), 5.11 (d, *J* = 12.0 Hz, 1H, PhCH_2_), 4.97 (d, *J* = 12.0 Hz, 1H, PhCH_2_), 2.56 (d, *J* = 13.8 Hz, 1H), 2.26–2.17 (m, 2H), 2.13–1.95 (m, 3H), 1.87–1.76 (m, 3H), 1.70–1.63 (m, 2H), 1.59–1.46 (m, 4H), 1.39–1.37 (m, 2H), 1.22–1.17 (m, 1H), 0.97–0.92 (m, 1H), 0.91 (s, 3H, H-16), 0.90 (s, 3H, H-17), 0.87 (s, 3H, H-19), 0.63 (s, 3H, H-18); ^13^C NMR (150 MHz, CDCl_3_): *δ* 175.3 (C-20), 163.3 (C-Ph), 132.0 (C-Ph), 131.9 (C-9), 130.1 (C-Ph), 130.0 (C-8), 115.3 (C-Ph), 71.9 (C-13), 65.4 (PhCH_2_), 52.3 (C-5), 48.8 (C-10), 41.8 (C-3), 41.3 (C-14), 34.4 (C-1), 34.0 (C-4), 33.8 (C-15), 31.9 (C-12), 31.9 (C-7), 31.8 (C-18), 21.6 (C-11), 20.1 (C-2), 20.0 (C-19), 18.1 (C-6), 16.7 (C-16), 16.7 (C-17); HRMS *m/z* calcd for C_27_H_37_O_3_FNa ([M + Na]^+^) 451.2619, found 451.2617 (Appendix A).

Data for **3g**: White solid, yield: 98%, m.p. 76–78 °C; IR cm^−1^: 1720, 1456, 1201, 1146, 1128, 1089, 981, 806; ^1^H NMR (600 MHz, CDCl_3_): *δ* 7.31 (d, *J* = 8.4 Hz, 2H, H-Ph), 7.25 (d, *J* = 8.4 Hz, 2H, H-Ph), 5.11 (d, *J* = 12.6 Hz, 1H, PhCH_2_), 4.95 (d, *J* = 12.6 Hz, 1H, PhCH_2_), 2.56 (d, *J* = 12.6 Hz, 1H), 2.26–2.17 (m, 2H), 2.13–1.96 (m, 3H), 1.85–1.76 (m, 3H), 1.70–1.46 (m, 6H), 1.40–1.37 (m, 2H), 1.22–1.17 (m, 1H), 0.97–0.92 (m, 1H), 0.91 (s, 3H, H-16), 0.90 (s, 3H, H-17), 0.88 (s, 3H, H-19), 0.64 (s, 3H, H-18); ^13^C NMR (150 MHz, CDCl_3_): *δ* 175.4 (C-20), 134.8 (C-Ph), 133.9 (C-Ph), 130.1 (C-9), 129.9 (C-Ph), 129.5 (C-8), 128.8 (C-Ph), 71.9 (C-13), 65.4 (PhCH_2_), 52.4 (C-5), 48.9 (C-10), 41.9 (C-3), 41.4 (C-14), 34.5 (C-1), 34.1 (C-4), 33.9 (C-15), 32.0 (C-12), 32.0 (C-7), 32.0 (C-18), 21.8 (C-11), 20.2 (C-2), 20.1 (C-19), 18.3 (C-6), 16.8 (C-16), 16.8 (C-17); HRMS *m/z* calcd for C_27_H_37_O_3_ClNa ([M + Na]^+^) 467.2323, found 467.2326 (Appendix A).

Data for **3h:** colorless oil, yield: 88%; IR cm^−1^: 1716, 1472, 1205, 1128; ^1^H NMR (600 MHz, CDCl_3_): *δ* 7.41–7.40 (m, 2H, H-Ph), 7.14–7.13 (m, 1H, H-Ph), 5.14 (d, *J* = 12.6 Hz, 1H, PhCH_2_), 4.89 (d, *J* = 12.6 Hz, 1H, PhCH_2_), 2.57 (d, *J* = 13.2 Hz, 1H), 2.22–2.16 (m, 2H), 2.14–1.98 (m, 3H), 1.88–1.77 (m, 3H), 1.73–1.48 (m, 6H), 1.42–1.39 (m, 2H), 1.23–1.18 (m, 1H), 0.99–0.94 (m, 1H), 0.92 (s, 3H, H-16), 0.90 (s, 3H, H-17), 0.89 (s, 3H, H-19), 0.66 (s, 3H, H-18); ^13^C NMR (150 MHz, CDCl_3_): *δ* 175.4 (C-20), 136.7 (C-Ph), 132.7 (C-Ph), 132.1 (C-Ph), 130.6 (C-9), 130.2 (C-Ph), 129.9 (C-8), 129.8 (C-Ph), 127.2 (C-Ph), 72.0 (C-13), 64.6 (PhCH_2_), 52.4 (C-5), 49.0 (C-10), 41.9 (C-3), 41.5 (C-14), 34.6 (C-1), 34.3 (C-4), 33.9 (C-15), 32.0 (C-12), 32.0 (C-7), 29.7 (C-18), 21.8 (C-11), 20.2 (C-2), 20.1 (C-19), 18.4 (C-6), 16.8 (C-16), 16.8 (C-17); HRMS *m/z* calcd for C_27_H_36_O_3_Cl_2_Na ([M + Na]^+^) 501.1934, found 501.1935 (Appendix A).

Data for **3i**: White solid, yield: 96%, m.p. 65–67 °C; IR cm^−1^: 1720, 1199, 1145, 1129, 1112, 746; ^1^H NMR (600 MHz, CDCl_3_): *δ* 7.55–7.52 (m, 1H, H-Ph), 7.37–7.28 (m, 2H, H-Ph), 7.17–7.15 (m, 1H, H-Ph), 5.28 (d, *J* = 13.2 Hz, 1H, PhCH_2_), 5.03 (d, *J* = 13.2 Hz, 1H, PhCH_2_), 2.59 (d, *J* = 13.8 Hz, 1H), 2.30–2.17 (m, 2H), 2.11–1.98 (m, 3H), 1.88–1.76 (m, 3H), 1.71–1.57 (m, 4H), 1.47–1.38 (m, 4H), 1.23–1.18 (m, 1H), 0.97–0.92 (m, 1H), 0.90 (s, 3H, H-16), 0.89 (s, 3H, H-17), 0.88 (s, 3H, H-19), 0.67 (s, 3H, H-18); ^13^C NMR (150 MHz, CDCl_3_): *δ* 175.3 (C-20), 135.6 (C-Ph), 132.8 (C-Ph), 132.6 (C-Ph), 130.2 (C-9), 129.8 (C-Ph), 129.7 (C-8), 127.7 (C-Ph), 123.7 (C-Ph), 71.9 (C-13), 65.7 (PhCH_2_), 52.4 (C-5), 49.0 (C-10), 42.0 (C-3), 41.5 (C-14), 34.6 (C-1), 33.9 (C-4), 33.8 (C-15), 32.1 (C-12), 32.0 (C-7), 32.0 (C-18), 21.8 (C-11), 20.2 (C-2), 20.1 (C-19), 18.3 (C-6), 16.8 (C-16), 16.8 (C-17); HRMS *m/z* calcd for C_27_H_37_O_3_BrNa ([M + Na]^+^): 511.1818, found 511.1822 (Appendix A).

Data for **3j**: White solid, yield: 93%, m.p. 63–65 °C; IR cm^−1^: 1719, 1455, 1200, 1146, 1127; ^1^H NMR (600 MHz, CDCl_3_): *δ* 7.47 (d, *J* = 8.4 Hz, 2H, H-Ph), 7.19 (d, *J* = 8.4 Hz, 2H, H-Ph), 5.10 (d, *J* = 12.6 Hz, 1H, PhCH_2_), 4.93 (d, *J* = 12.6 Hz, 1H, PhCH_2_), 2.57 (d, *J* = 13.2 Hz, 1H), 2.25–2.17 (m, 2H), 2.13–1.96 (m, 3H), 1.87–1.77 (m, 3H), 1.71–1.47 (m, 6H), 1.40–1.37 (m, 2H), 1.22–1.17 (m, 1H), 0.98–0.92 (m, 1H), 0.91 (s, 3H, H-16), 0.90 (s, 3H, H-17), 0.88 (s, 3H, H-19), 0.65 (s, 3H, H-18); ^13^C NMR (150 MHz, CDCl_3_): *δ* 175.4 (C-20), 135.3 (C-Ph), 131.7 (C-Ph), 130.1 (C-9), 130.0 (C-Ph), 129.9 (C-8), 122.1 (C-Ph), 72.0 (C-13), 65.5 (PhCH_2_), 52.5 (C-5), 49.0 (C-10), 41.9 (C-3), 41.5 (C-14), 34.6 (C-1), 34.2 (C-4), 33.9 (C-15), 32.1 (C-12), 32.0 (C-7), 32.0 (C-18), 21.8 (C-11), 20.2 (C-2), 20.2 (C-19), 18.3 (C-6), 16.8 (C-16), 16.8 (C-17); HRMS *m/z* calcd for C_27_H_37_O_3_BrNa ([M + Na]^+^) 511.1818, found 511.1822 (Appendix A).

Data for **3k**: colorless oil, yield: 95%; IR cm^−1^: 1714, 1455, 1202, 1126; ^1^H NMR (600 MHz, CDCl_3_): *δ* 7.09 (d, *J* = 7.8 Hz, 1H, H-Ph), 7.06 (s, 1H, H-Ph), 7.03 (d, *J* = 7.2 Hz, 1H, H-Ph), 5.09 (d, *J* = 12.6 Hz, 1H, PhCH_2_), 4.92 (d, *J* = 12.0 Hz, 1H, PhCH_2_), 2.57 (d, *J* = 13.2 Hz, 1H), 2.30–2.17 (m, 8H), 2.13–1.96 (m, 3H), 1.91–1.78 (m, 3H), 1.69–1.63 (m, 2H), 1.59–1.48 (m, 3H), 1.39–1.38 (m, 3H), 1.22–1.17 (m, 1H), 0.96–0.91 (m, 1H), 0.90 (s, 3H, H-16), 0.89 (s, 3H, H-17), 0.88 (s, 3H, H-19), 0.68 (s, 3H, H-18); ^13^C NMR (150 MHz, CDCl_3_): *δ* 175.5 (C-20), 136.7 (C-Ph), 136.4 (C-Ph), 133.7 (C-Ph), 130.2 (C-9), 129.8 (C-Ph), 129.7 (C-8), 129.4 (C-Ph), 125.5 (C-Ph), 71.9 (C-13), 66.2 (PhCH_2_), 52.4 (C-5), 48.9 (C-10), 42.0(C-3), 41.5 (C-14), 34.5 (C-1), 34.0 (C-4), 33.9 (C-15), 32.1 (C-12), 32.1 (C-7), 32.0 (C-18), 21.8 (C-11), 20.2 (C-2), 20.2 (C-19), 19.8 (CH_3_Ph), 19.6 (CH_3_Ph), 18.3 (C-6), 16.8 (C-16), 16.8 (C-17); HRMS *m/z* calcd for C_29_H_42_O_3_Na([M + Na]^+^) 461.3026, found 461.3025 (Appendix A).

Data for **3l**: colorless oil, yield: 94%; IR cm^−1^: 1713, 1515, 1249; ^1^H NMR (600 MHz, CDCl_3_): *δ* 7.25 (d, *J* = 8.0 Hz, 2H, H-Ph), 6.86 (d, *J* = 8.0 Hz, 2H, H-Ph), 5.06 (d, *J* = 12.0 Hz, 1H, PhCH_2_), 4.95 (d, *J* = 12.0 Hz, 1H, PhCH_2_), 3.79 (s, 3H, OCH_3_), 2.55 (d, *J* = 13.2 Hz, 1H), 2.28–2.16 (m, 2H), 2.12–1.96 (m, 3H), 1.88–1.76 (m, 3H), 1.68–1.63 (m, 2H), 1.58–1.45 (m, 4H), 1.39–1.35 (m, 2H), 1.21–1.16 (m, 1H), 0.95–0.92 (m, 1H), 0.90 (s, 3H, H-16), 0.89 (s, 3H, H-17), 0.87 (s, 3H, H-19), 0.65 (s, 3H, H-18); ^13^C NMR (150 MHz, CDCl_3_): *δ* 175.5 (C-20), 159.5 (C-Ph), 130.1 (C-9), 129.9 (C-Ph), 129.8 (C-8), 128.3 (C-Ph), 113.9 (C-Ph), 71.9 (C-13), 66.0 (PhCH_2_), 55.3 (OCH_3_), 52.4 (C-5), 48.9 (C-10), 41.9 (C-3), 41.4 (C-14), 34.5 (C-1), 34.1 (C-4), 33.9 (C-15), 32.1 (C-12), 32.0 (C-7), 31.9 (C-18), 21.7 (C-11), 20.2 (C-2), 20.1 (C-19), 18.2 (C-6), 16.8 (C-16), 16.8 (C-17); HRMS *m/z* calcd for C_28_H_40_O_4_Na ([M + Na]^+^) 463.2819, found 463.2820 (Appendix A).

Data for **3m**: White solid, yield: 91%, m.p. 163–165 °C; IR cm^−1^: 1715, 1525, 1358, 1199, 736; ^1^H NMR (600 MHz, CDCl_3_):*δ* 8.06 (d, *J* = 7.8 Hz, 1H, H-Ph), 7.60 (td, *J* = 7.8, 1.2 Hz, 1H, H-Ph), 7.52 (d, *J* = 9.0 Hz, 1H, H-Ph), 7.48 (t, *J* = 7.2 Hz, 1H, H-Ph), 5.72 (d, *J* = 14.4 Hz, 1H, PhCH_2_), 5.21 (d, *J* = 14.4 Hz, 1H, PhCH_2_), 2.60 (d, *J* = 13.2 Hz, 1H), 2.23–1.98 (m, 5H), 1.91–1.78 (m, 3H), 1.72–1.64 (m, 2H), 1.60–1.51 (m, 4H), 1.42–1.40 (m, 2H), 1.24–1.18 (m, 1H), 0.99–0.94 (m, 1H), 0.93 (s, 3H, H-16), 0.92 (s, 3H, H-17), 0.88 (s, 3H, H-19), 0.69 (s, 3H, H-18); ^13^C NMR (150 MHz, CDCl_3_):*δ* 175.1 (C-20), 147.8 (C-Ph), 133.9 (C-Ph), 132.5 (C-Ph), 130.6 (C-9) 129.7 (C-Ph), 129.5 (C-8), 128.9 (C-Ph), 125.1 (C-Ph), 71.9 (C-13), 63.0 (PhCH_2_), 52.3 (C-5), 49.0 (C-10), 42.0 (C-3), 41.5 (C-14), 34.8 (C-1), 34.5 (C-4), 34.0 (C-15), 32.2 (C-12), 32.1 (C-7), 31.9 (C-18), 21.5 (C-11), 20.1 (C-2), 20.1 (C-19), 18.4 (C-6), 16.9 (C-16), 16.8 (C-17); HRMS *m/z* calcd for C_27_H_37_O_5_NNa([M + Na]^+^) 478.2564, found 478.2568 (Appendix A).

Data for **3n**: White solid, yield: 95%, m.p. 99–101 °C; IR cm^−1^: 1710, 1518, 1345, 1162, 1122; ^1^H NMR (600 MHz, CDCl_3_): *δ* 8.21 (d, *J* = 8.4 Hz, 2H, H-Ph), 7.48 (d, *J* = 8.4 Hz, 2H, H-Ph), 5.29 (d, *J* = 13.2 Hz, 1H, PhCH_2_), 5.05 (d, *J* = 13.5 Hz, 1H, PhCH_2_), 2.60 (d, *J* = 13.2 Hz, 1H), 2.26–2.19 (m, 2H), 2.13–1.99 (m, 3H), 1.87–1.77 (m, 3H), 1.75–1.45 (m, 6H), 1.44–1.39 (m, 2H), 1.25–1.19 (m, 1H), 1.01–0.96 (m, 1H), 0.92 (s, 3H, H-16), 0.91 (s, 3H, H-17), 0.89 (s, 3H, H-19), 0.66 (s, 3H, H-18); ^13^C NMR (150 MHz, CDCl_3_): *δ* 175.3 (C-20), 147.7 (C-Ph), 143.7 (C-Ph), 130.4 (C-9), 129.7 (C-8), 128.3 (C-Ph), 123.9 (C-Ph), 72.0 (C-13), 64.8 (PhCH_2_), 52.4 (C-5), 49.0 (C-10), 41.8 (C-3), 41.4 (C-14), 34.6 (C-1), 34.6 (C-4), 33.9 (C-15), 32.0 (C-12), 32.0 (C-7), 32.0 (C-18), 21.7 (C-11), 20.2 (C-2), 20.1 (C-19), 18.3 (C-6), 16.8 (C-16), 16.8 (C-17); HRMS *m/z* calcd for C_27_H_37_O_5_NNa ([M + Na]^+^) 478.2564, found 478.2568 (Appendix A).

#### 3.1.2. Synthesis of Compound **4**

To a suspension of compound **3a** (100 mg, 0.30 mmol) in dry tetrahydrofuran (THF, 10 mL) at 0 °C under N_2_ was added lithium aluminum hydride (56 mg, 1.5 mmol) in dry THF (2 mL) dropwise over 10 min. The resulting mixture was allowed to heat to 40 °C for 1 h. When the reaction was complete, pure water (30 mL) was added to the reaction, The solvent was removed and the residue was diluted by ethyl acetate (30 mL), washed with saturated brine (30 mL), dried over anhydrous Na_2_SO_4_, concentrated in vacuo, and purified by silica gel column chromatography to afford **4** in 83%.

Data for **4**: White solid, yield: 83%, m.p. 168–170 °C; IR cm^−1^: 3294, 2943, 1362, 1046, 935; ^1^H NMR (600 MHz, CDCl_3_): *δ* 3.97 (d, *J* = 11.4 Hz, 1H, H-20), 3.53 (d, *J* = 11.4 Hz, 1H, H-20), 2.15–2.01 (m, 6H), 1.97–1.85 (m, 3H), 1.81–1.78 (m, 1H), 1.67–1.60 (m, 3H), 1.52–1.43 (m, 4H), 1.31 (dd, *J* = 13.2, 3.0 Hz, 1H), 1.20–1.11 (m, 2H), 0.94–0.92 (m, 9H, H-16, 17, 19), 0.90 (s, 3H, H-18); ^13^C NMR (150 MHz, CDCl_3_): *δ* 132.4 (C-9), 129.1 (C-8), 72.5 (C-13), 66.0 (C-20), 51.0 (C-5), 42.8 (C-10), 41.9 (C-3), 40.3 (C-14), 36.8 (C-1), 34.9 (C-4), 33.6 (C-15), 32.5 (C-12), 32.5 (C-7), 31.4 (C-18), 21.9 (C-11), 21.9 (C-2), 19.2 (C-19), 18.8 (C-6), 17.0 (C-16), 16.9 (C-17); HRMS *m/z* calcd for C_20_H_34_O_2_Na p([M + Na]^+^) 329.2451, found 329.2451 (Appendix A).

#### 3.1.3. General Procedure for the Synthesis of Compound **5a–i**

To a solution of **4** (100 mg, 0.33 mmol) and Sodium hydride (26 mg, 1.0 mmol) in DMF (10 mL) was added substituent haloalkane (0.66 mmol) in DMF (2 mL) was added dropwise for 10 min. The resulting mixture was stirred at room temperature for 1.5–2 h until the starting materials were completely transformed. After termination by pure water (30 mL), the reaction was extracted with ethyl acetate (3 × 30 mL). The combined organic phase was dried over anhydrous Na_2_SO_4_, filtered, concentrated under reduced pressure, and purified by silica gel column chromatography eluting with petroleum ether/ethyl acetate to afford compound **5a-i** in 42–74%.

Data for **5a:** colorless oil, yield: 43%; IR cm^−1^: 2925, 2358, 988, 957; ^1^H NMR (600 MHz, CDCl_3_): *δ* 3.51–3.48 (m, 2H, H-20), 3.25 (s, 3H, OCH_3_), 2.24–2.05 (m, 6H), 1.94–1.90 (m, 1H), 1.74–1.58 (m, 5H), 1.52–1.42 (m, 4H), 1.30–1.27 (m, 1H), 1.20–1.15 (m, 1H), 1.11–1.08 (m, 1H), 0.95–0.92 (m, 6H, H-16, 17), 0.89 (s, 3H, H-19), 0.86 (s, 3H, H-18); ^13^C NMR (150 MHz, CDCl_3_): *δ* 134.9 (C-9), 126.2 (C-8), 76.8 (C-20), 72.3 (C-13), 59.3 (OCH_3_), 51.1 (C-5), 41.9 (C-10), 41.6 (C-3), 40.3 (C-14), 36.1 (C-1), 34.2 (C-4), 33.5 (C-15), 33.4 (C-12), 32.6 (C-7), 31.1 (C-18), 22.9 (C-11), 22.1 (C-2), 19.2 (C-19), 18.6 (C-6), 17.2 (C-16), 17.2 (C-17); HRMS *m/z* calcd for C_21_H_36_O_2_Na ([M + Na]^+^): 343.2608, found 343.2607 (Appendix A).

Data for **5b:** colorless oil, yield: 73%; IR cm^−1^: 2925, 2358, 988, 957; ^1^H NMR (600 MHz, CDCl_3_): *δ* 3.53 (s, 2H, H-20), 3.37 (q, *J* = 7.2 Hz, 2H, CH_2_CH_3_), 2.30–2.06 (m, 6H), 1.94–1.90 (m, 1H), 1.74–1.65 (m, 4H), 1.60–1.56 (m, 1H), 1.53–1.41 (m, 4H), 1.30–1.28 (m, 1H), 1.20–1.15 (m, 1H), 1.14 (t, *J* = 7.2 Hz, 3H, CH_2_CH_3_), 1.10–1.06 (m, 1H), 0.95–0.92 (m, 6H, H-16, 17), 0.89 (s, 3H, H-19), 0.85 (s, 3H, H-18); ^13^C NMR (150 MHz, CDCl_3_): *δ* 134.9 (C-9), 126.1 (C-8), 74.4 (C-20), 72.3 (C-13), 66.7 (CH_2_CH_3_), 51.0 (C-5), 42.0 (C-10), 41.5 (C-3), 40.2 (C-14), 36.2 (C-1), 34.2 (C-4), 33.5 (C-15), 33.4 (C-12), 32.5 (C-7), 31.1 (C-18), 22.9 (C-11), 22.1 (C-2), 19.2 (C-19), 18.6 (C-6), 17.2 (C-16), 17.2 (C-17), 15.4 (CH_2_CH_3_); HRMS *m/z* calcd for C_22_H_38_O_2_Na ([M + Na]^+^): 357.2764, found 357.2764 (Appendix A).

Data for **5c:** colorless oil, yield: 43%; IR cm^−1^: 2941, 1455, 1386, 1090; ^1^H NMR (600 MHz, CDCl_3_): *δ* 7.33–7.23(m, 5H, H-Ph), 4.47 (d, *J* = 12.6 Hz, 1H, PhCH_2_), 4.41 (d, *J* = 12.6 Hz, 1H, PhCH_2_), 3.59–3.55 (m, 2H, H-20), 2.34–2.02 (m, 6H), 1.95–1.91 (m, 1H), 1.76–1.58 (m, 5H), 1.51–1.37 (m, 4H), 1.28–1.25 (m, 1H), 1.17–1.12 (m, 1H), 1.07–1.02 (m, 1H), 0.94–0.92 (m, 6H, H-16, 17), 0.86 (s, 3H, H-19), 0.74 (s, 3H, H-18); ^13^C NMR (150 MHz, CDCl_3_):*δ* 138.9 (C-Ph), 135.1 (C-Ph), 128.4 (C-9), 127.6 (C-8), 127.4 (C-Ph), 126.1 (C-Ph), 74.0 (C-20), 73.5 (PhCH_2_), 72.3 (C-13), 51.2 (C-5), 41.9 (C-10), 41.5 (C-3), 40.4 (C-14), 35.8 (C-1), 34.0 (C-4), 33.5 (C-15), 33.5 (C-12), 32.4 (C-7), 31.4 (C-18), 22.8 (C-11), 22.0 (C-2), 19.1 (C-19), 18.6 (C-6), 17.2 (C-16), 17.1 (C-17); HRMS *m/z* calcd for C_27_H_40_O_2_Na ([M + Na]^+^) 419.2921, found 419.2913 (Appendix A).

Data for **5d:** colorless oil, yield: 46%; IR cm^−1^: 2941, 1455, 1386, 1090; ^1^H NMR (600 MHz, CDCl_3_): *δ* 7.26–7.23 (m, 2H, H-Ph), 7.02–6.99 (m, 2H, H-Ph), 4.43 (d, *J* = 12.0 Hz, 1H, PhCH_2_), 4.36 (d, *J* = 12.0 Hz, 1H, PhCH_2_), 3.57–3.53 (m, 2H, H-20), 2.31–2.04 (m, 6H), 1.94–1.91 (m, 1H), 1.75–1.57 (m, 5H), 1.51–1.38 (m, 4H), 1.28–1.25 (m, 1H), 1.18–1.13 (m, 1H), 1.08–1.03 (m, 1H), 0.94–0.92 (m, 6H, H-16, 17), 0.87 (s, 3H, H-19), 0.75 (s, 3H, H-18); ^13^C NMR (150 MHz, CDCl_3_): *δ* 163.1 (C-Ph), 135.0 (C-Ph), 129.4 (C-9), 129.3 (C-8), 126.2 (C-Ph), 115.3 (C-Ph), 74.05 (C-20), 72.8 (PhCH_2_), 72.3 (C-13), 51.2 (C-5), 41.9 (C-10), 41.5 (C-3), 40.4 (C-14), 35.9 (C-1), 34.0 (C-4), 33.5 (C-15), 33.5 (C-12), 32.4 (C-7), 31.4 (C-18), 22.8 (C-11), 22.0 (C-2), 19.1 (C-19), 18.6 (C-6), 17.2 (C-16), 17.1 (C-17); HRMS *m/z* calcd for C_27_H_39_O_2_FNa ([M + Na]^+^) 437.2828, found 437.2823 (Appendix A).

Data for **5e**: colorless oil, yield: 53%; IR cm^−1^: 2921, 1490, 1455, 1089, 1015, 806; ^1^H NMR (600 MHz, CDCl_3_): *δ* 7.29 (d, *J* = 7.8 Hz, 2H, H-Ph), 7.22 (d, *J* = 8.4 Hz, 2H, H-Ph), 4.43 (d, *J* = 12.0 Hz, 1H, PhCH_2_), 4.36 (d, *J* = 12.0 Hz, 1H, PhCH_2_), 3.59–3.54 (m, 2H, H-20), 2.31–2.04 (m, 6H), 1.95–1.91 (m, 1H), 1.76–1.57 (m, 5H), 1.51–1.39 (m, 4H), 1.29–1.26 (m, 1H), 1.18–1.13 (m, 1H), 1.09–1.03 (m, 1H), 0.94–0.92 (m, 6H, H-16, 17), 0.87 (s, 3H, H-19), 0.76 (s, 3H, H-18); ^13^C NMR (150 MHz, CDCl_3_): *δ* 137.4 (C-Ph), 135.0 (C-Ph), 133.2 (C-Ph), 128.9 (C-9), 128.6 (C-8), 126.3 (C-Ph), 74.3 (C-20), 72.8 (PhCH_2_), 72.3 (C-13), 51.2 (C-5), 41.8 (C-10), 41.5 (C-3), 40.4 (C-14), 35.9 (C-1), 34.0 (C-4), 33.5 (C-15), 32.4 (C-12), 31.4 (C-7), 31.4 (C-18), 22.8 (C-11), 22.0 (C-2), 19.1 (C-19), 18.6 (C-6), 17.1 (C-16), 17.1 (C-17); HRMS *m/z* calcd for C_27_H_39_O_2_ClNa ([M + Na]^+^) 453.2531, found 453.2533 (Appendix A).

Data for **5f**: white solid, yield: 53%, m.p. 63–65 °C; IR cm^−1^: 2358, 1455, 1089, 1012, 800, 668; ^1^H NMR (600 MHz, CDCl_3_): *δ* 7.44 (d, *J* = 8.4 Hz, 2H, H-Ph), 7.16 (d, *J* = 8.4 Hz, 2H, H-Ph), 4.42 (d, *J* = 12.6 Hz, 1H, PhCH_2_), 4.35 (d, *J* = 12.6 Hz, 1H, PhCH_2_), 3.59–3.54 (m, 2H, H-20), 2.29–2.03 (m, 6H), 1.95–1.91 (m, 1H), 1.76–1.57 (m, 5H), 1.51–1.39 (m, 4H), 1.29–1.25 (m, 1H), 1.18–1.13 (m, 1H), 1.08–1.03 (m, 1H), 0.94–0.91 (m, 6H, H-16, 17), 0.87 (s, 3H, H-19), 0.76 (s, 3H, H-18); ^13^C NMR (150 MHz, CDCl_3_): *δ* 137.9 (C-Ph), 135.0 (C-Ph), 131.5 (C-9), 129.2 (C-8), 126.2 (C-Ph), 121.3 (C-Ph), 74.3 (C-20), 72.8 (PhCH_2_), 72.2 (C-13), 51.2 (C-5), 41.8 (C-10), 41.5 (C-3), 40.4 (C-14), 35.9 (C-1), 33.9 (C-4), 33.5 (C-15), 33.4 (C-12), 32.4 (C-7), 31.4 (C-18), 22.8 (C-11), 22.0 (C-2), 19.1 (C-19), 18.6 (C-6), 17.1 (C-16), 17.1 (C-17); HRMS *m/z* calcd for C_27_H_39_O_2_BrNa ([M + Na]^+^) 497.2026, found 497.2028 (Appendix A).

Data for **5g**: colorless oil, yield: 74%; IR cm^−1^: 2928, 1466, 1091; ^1^H NMR (600 MHz, CDCl_3_): *δ* 7.16 (d, *J* = 7.8 Hz, 2H, H-Ph), 7.12 (d, *J* = 7.8 Hz, 2H, H-Ph), 4.43 (d, *J* = 12.0 Hz, 1H, PhCH_2_), 4.36 (d, *J* = 12.0 Hz, 1H, PhCH_2_), 3.57–3.54 (m, 2H, H-20), 2.32 (s, 3H, CH_3_), 2.29–2.03 (m, 6H), 1.94–1.91 (m, 1H), 1.75–1.57 (m, 5H), 1.50–1.37 (m, 4H), 1.27–1.25 (m, 1H), 1.17–1.12 (m, 1H), 1.08–1.01 (m, 1H), 0.94–0.91 (m, 6H, H-16, 17), 0.86 (s, 3H, H-19), 0.76 (s, 3H, H-18); ^13^C NMR (150 MHz, CDCl_3_): *δ* 137.0 (C-Ph), 135.1 (C-Ph), 129.1 (C-Ph), 129.0 (C-9), 128.0 (C-Ph), 127.6 (C-8), 73.9 (C-20), 73.3 (PhCH_2_), 72.2 (C-13), 51.2 (C-5), 41.9 (C-10), 41.5 (C-3), 40.4 (C-14), 35.8 (C-1), 34.0 (C-4), 33.5 (C-15), 33.4 (C-12), 32.4 (C-7), 31.4 (C-18), 22.7 (C-11), 22.0 (C-2), 21.2 (CH_3_), 19.1 (C-19), 18.6 (C-6), 17.1 (C-16), 17.1 (C-17); HRMS *m/z* calcd for C_28_H_42_O_2_Na ([M + Na]^+^) 433.3077, found 433.3076 (Appendix A).

Data for **5h**: colorless oil, yield: 43%; IR cm^−1^: 2941, 1455, 1386, 1090; ^1^H NMR (600 MHz, CDCl_3_): *δ* 7.07 (d, *J* = 7.2 Hz, 1H, H-Ph), 7.04 (s, 1H, H-Ph), 7.00 (d, *J* = 7.2 Hz, 1H, H-Ph), 4.41 (d, *J* = 12.1 Hz, 1H, PhCH_2_), 4.34 (d, *J* = 12.0 Hz, 1H, PhCH_2_), 3.57–3.54 (m, 2H, H-20), 2.34–2.27 (m, 1H), 2.24 (s, 3H, CH_3_), 2.23 (s, 3H, CH_3_), 2.18–2.04 (m, 5H), 1.95–1.91 (m, 1H), 1.76–1.58 (m, 5H), 1.51–1.38 (m, 4H), 1.28–1.25 (m, 1H), 1.17–1.12 (m, 1H), 1.07–1.02 (m, 1H), 0.95–0.92 (m, 6H, H-16, 17), 0.86 (s, 3H, H-19), 0.76 (s, 3H, H-18); ^13^C NMR (150 MHz, CDCl_3_): *δ* 136.3 (C-Ph), 135.6 (C-Ph), 135.1 (C-Ph), 129.6 (C-9), 129.0 (C-Ph), 126.1 (C-Ph), 125.1 (C-8), 73.9 (C-20), 73.3 (PhCH_2_), 72.2 (C-13), 51.2 (C-5), 41.9 (C-10), 41.5 (C-3), 40.5 (C-14), 35.8 (C-1), 34.1 (C-4), 33.5 (C-15), 33.5 (C-12), 32.5 (C-7), 31.4 (C-18), 22.8 (C-11), 22.0 (C-2), 19.8 (C-19), 19.6 (CH_3_), 19.1 (CH_3_), 18.6 (C-6), 17.2 (C-16), 17.1 (C-17); HRMS *m/z* calcd for C_29_H_44_O_2_Na ([M + Na]^+^) 447.3234, found 447.3233 (Appendix A).

#### 3.1.4. Synthesis of Compound **6a**–**c**

To a solution of **3a** or **3d** (0.3 mmol) and pyridine (1.2 mmol) in DMF (10 mL) at 0 °C was added acetyl chloride (0.6 mmol). The mixture was warmed up to room temperature and stirred for 2–2.5 h. When the reaction was complete, pure water (30 mL) was added to the reaction, which was extracted with ethyl acetate (3 × 30 mL). The combined organic phase was dried over anhydrous Na_2_SO_4_, filtered, concentrated under reduced pressure, and purified by silica gel column chromatography eluting with petroleum ether/ethyl acetate to afford compounds **6a** (75%), **6b** (65%), or **6c** (15%).

Data for **6a:** white solid, yield: 75%, m.p. 118–120 °C; IR cm^−1^: 1726, 1361, 1272, 1226, 1203, 1130; ^1^H NMR (600 MHz, CDCl_3_): *δ* 3.59 (s, 3H, OCH_3_), 2.57–2.52 (m, 2H), 2.39 (d, *J* = 17.4 Hz, 1H), 2.34–2.30 (m, 1H), 2.28–2.24 (m, 1H), 2.21–1.97 (m, 4H), 1.90 (s, 3H, COCH_3_), 1.84–1.76 (m, 1H), 1.74–1.68 (m, 2H), 1.62–1.57 (m, 1H), 1.55–1.51 (m, 1H), 1.40–1.35 (m, 2H), 1.22–1.17 (m, 1H), 0.95–0.90 (m, 1H), 0.89–0.88 (m, 9H, H-16, 17, 19), 0.68 (s, 3H, H-18); ^13^C NMR (150 MHz, CDCl_3_): *δ* 176.1 (C-20), 170.7 (COCH_3_), 129.6 (C-9), 129.3 (C-8), 85.9 (C-13), 52.4 (OCH_3_), 51.3 (C-5), 48.6 (C-10), 42.0 (C-3), 36.2 (C-14), 34.8 (C-1), 33.9 (C-4), 32.1 (C-15), 32.1 (C-12), 32.1 (C-7), 31.8 (C-18), 22.3 (COCH_3_), 21.7 (C-11), 20.3 (C-2), 20.1 (C-19), 18.3 (C-6), 17.5 (C-16), 17.2 (C-17); HRMS *m/z* calcd for C_23_H_36_O_4_Na ([M + Na]^+^) 399.2506, found 399.2507 (Appendix A).

Data for **6b:** colorless oil, yield: 65%; IR cm^−1^: 2957, 1748, 1687, 1458, 1127, 906; ^1^H NMR (600 MHz, CDCl_3_): *δ* 7.36–7.28 (5H, H-Ph), 5.13 (d, *J* = 12.6 Hz, 1H, PhCH_2_), 4.98 (d, *J* = 12.6 Hz, 1H, PhCH_2_), 2.61 (d, *J* = 16.8 Hz, 1H), 2.54–2.50 (m, 1H), 2.43 (d, *J* = 12.6 Hz, 1H), 2.35–2.27 (m, 2H), 2.23 (d, *J* = 16.8 Hz, 1H), 2.17–2.13 (m, 1H), 2.10–1.98 (m, 2H), 1.85–1.63 (m, 6H), 1.64–1.59 (m, 1H), 1.54–1.52 (m, 1H), 1.40–1.37 (m, 2H), 1.22–1.17 (m, 1H), 0.97–0.92 (m, 1H), 0.89–0.87 (m, 9H, H-16, 17, 19), 0.66 (s, 3H, H-18); ^13^C NMR (150 MHz, CDCl_3_): *δ* 175.3 (C-20), 170.7 (COCH_3_), 136.3 (C-Ph), 129.8 (C-9), 129.6 (C-8), 128.6 × 2 (C-Ph), 128.0 (C-Ph), 127.8 (C-Ph) × 2, 85.9 (C-13), 66.2 (PhCH_2_), 52.3 (C-5), 48.8 (C-10), 42.0 (C-3), 36.3 (C-14), 34.7 (C-1), 34.0 (C-4), 32.0 (C-15), 31.9 (C-12), 31.6 (C-7), 31.6 (C-18), 28.2 (COCH_3_), 22.3 (C-11), 21.7 (C-2), 20.2 (C-19), 18.3 (C-6), 17.5 (C-16), 17.2 (C-17); HRMS *m/z* calcd for C_29_H_40_O_4_Na ([M + Na]^+^): 475.2810, found 475.2811 (Appendix A).

Data for **6c:** colorless oil, yield: 15%; IR cm^−1^: 2359, 1716, 1704, 1206; ^1^H NMR (600 MHz, CDCl_3_): *δ* 3.62 (s, 3H, OCH_3_), 3.33–3.27 (m, 2H, COCH_2_COCH_3_), 2.57 (d, *J* = 13.8 Hz, 1H), 2.52–2.48 (m, 1H), 2.43 (d, *J* = 16.8 Hz, 1H), 2.36–2.15 (m, 7H), 2.09–1.97 (m, 2H), 1.83–1.63 (m, 4H), 1.54–1.50 (m, 1H), 1.40–1.33 (m, 2H), 1.22–1.17 (m, 1H), 0.95–0.92 (m, 1H), 0.91–0.89 (m, 9H, H-16, 17, 19), 0.68 (s, 3H, H-18); ^13^C NMR (150 MHz, CDCl_3_):*δ* 201.2 (COCH_2_COCH_3_), 175.9 (C-20), 166.5 (COCH_2_COCH_3_), 129.9 (C-9), 129.2 (C-8), 88.0 (C-13), 52.5 (C-5), 51.4 (OCH_3_), 51.3 (COCH_2_COCH_3_), 48.6 (C-10), 42.0 (C-3), 36.2 (C-14), 34.8 (C-1), 33.8 (C-4), 32.1 (C-15), 32.1 (C-12), 31.8 (C-7), 30.2 (C-18), 28.2 (C-25), 21.7 (C-11), 20.3 (C-2), 20.1 (C-19), 18.2 (C-6), 17.5 (C-16), 17.2 (C-17);HRMS *m/z* calcd for C_25_H_38_O_5_Na ([M + Na]^+^): 441.2611, found 441.2612 (Appendix A).

### 3.2. Pathogens and Culture Conditions

The microbial pathogen strains used for bioassays were methicillin-resistant *Staphylococcus aureus* (ATCC 43300), *Escherichia coli* (ATCC 25922), *Streptococcus mutans* (ATCC 35668), *Streptococcus sobrinus* (ATCC 33478), *Candida albicans* (ATCC 10231), *Cryptococcus neoformans* (ATCC 66031), *T. rubrum* (ATCC-MYA4438) and *T. mentagrophyte* (ATCC9533). The bacteria strains *S. mutans* and *S. sobrinus* (ATCC 33478) were obtained from the Prince Philip Dental Hospital, Hong Kong University, Hong Kong.

The strains of *S. aureus* and *E. coli* were grown at 37 °C on Tryptic Soy media (TSA, TSB; BD Biosciences, CA, USA). Yeast malt (YM; BD Biosciences, CA, USA) media were used for cultivating *C. albicans* at 30 °C. *T. rubrum, T. mentagrophyte,* and *C. neoformans* were grown at 30 °C on Sabouraud dextrose media (SDA; BD Biosciences). *S. mutans* and *S. sobrinus* were started from the frozen BHI-glycerol stocks and growth at 37 °C with shaking. Strains grown in liquid media were cultivated on an orbital shaker at 200 rpm.

### 3.3. In Vitro Antifungal Assay

To evaluate the antifungal activity against filamentous fungi, *T. rubrum* and *T. mentagrophytes* were separately cultured for 2 weeks at 28 °C on SDA to produce conidia. A mixed suspension of conidia and hyphae fragments was obtained by covering the fungal colonies with sterile saline (0.85%) and gently rubbing the colonies with the inoculation loop. Then, the suspension was filtered with four layers of sterile lens paper to remove the hyphae and centrifuged at 1000× *g* for 10 min to collect the conidia. Conidia were washed twice by agitation in sterile saline. The concentration of conidia or spore was adjusted with sterile saline to 1 × 10^4^ cells/mL by hemocytometer counts. The antifungal susceptibility testing was performed as outlined in document M38-A2 and previous research, with minor changes [16,17,18,19]. The medium used was RPMI 1640 with L-glutamine buffered to pH 7.0 with 0.165 M morpholinepropanesulfonic acid (MOPS) and was supplemented with 2% glucose (*m*/*v*). The 195 μL of prepared conidia or spore suspension was seeded on 96-well plates that had been previously added with 5 μL of tested agents in each well, and three replicates were used for each treatment. Miconazole was used as a positive control in the assay. The 96-well plates were then incubated at 28 ± 2 °C for 7 days. The optical density (OD) reading was measured by a microplate reader at 510 nm. The fungal growth inhibition is determined using the formula:(1)$$\mathrm{Inhibition}\%=\left(1-\frac{\mathrm{ODexperimental}-\mathrm{OD}\;\mathrm{blank}}{\mathrm{ODnegative}}\right)\times 100\%$$

Antifungal susceptibility testing of yeasts was performed by using a micro-broth dilution assay. The compounds were dissolved in DMSO at a stock concentration of 2–10 mg/mL and kept at 4 °C for the bioassays. Exponentially growing cultures of each strain were prepared from overnight cultures, and cultures were adjusted to the OD value of about 0.5 at 600 nm. Cultures were then diluted 1:1000 in broth (C. neoformans was directly used at OD600 = 0.05) and added to a 96-well plate (195 µL/well). Miconazole (Dalian Meilun, Dalian, China; 10 µg mL^−1^) was used as a positive control for C. albicans, and C. neoformans. Plates were read at 600 nm after incubation for 48 h. Inhibition was calculated by subtracting the absorbance of the blank wells, dividing by the average value for the DMSO-only wells, and multiplying by 100.

### 3.4. In Vitro Antibacterial Assay

Compounds were tested for planktonic microbial growth inhibition using the above micro-broth dilution assay [16]. The compounds and the standard drug were prepared in DMSO. Exponentially growing cultures of S. aureus, E. coli, S. mutans, and S. sobrinus were prepared from overnight cultures, and cultures were adjusted to the OD value of about 0.5 at 600 nm. Cultures were then diluted 1:1000 in broth and added to a 96-well plate (195 µL/well). Tetracycline (Sigma, St. Louis, MO, USA; 10 µg mL−1 in DMSO) was used as a standard drug. Plates were read at 600 nm after incubation for 24 h. Inhibition was calculated by the above calculation formula.

## 4. Conclusions

A series of novel lophanic acid derivatives have been prepared and evaluated for their antifungal and antibacterial activities. Among the derivatives, **3d** is the only compound that showed > 70% inhibitory effects against three fungal and bacterial strains (*T. mentagrophytes, C. neoformans*, and *MRSA*), and **3b**, **5c,** and **5h** were found to be able to inhibit the microbial growth of *MRSA* by over 90%. Through a structure-activity relationship analysis, we observed the presence of a C-20 carboxylic group and a free hydroxyl group at C-13 is essential to retain broad antimicrobial activities for the lophanic acid derivatives (e.g., **3a**, **3b**, **3d,** and **3i**). Without the C-20 carboxylic group, the inhibitory effects of the lophanic acid derivatives against *T. rubrum* and *C. neoformans* were much weakened (e.g., **5c** and **5h**). Our present study determined that the C-20 carboxylic group could be the key position for a structural modification to obtain lophanic acid analogs with broad-spectrum antimicrobial activity.

## Data Availability

Not applicable.

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
