# Peer review of "Design, Synthesis and Biological Evaluation of Lophanic Acid Derivatives as Antifungal and Antibacterial Agents"

_molecules, 2022, doi:10.3390/molecules27206836_

Round 1

Reviewer 1 Report

The article describes the synthesis and antimicrobial activity of new lophanic acid derivatives. The work is well designed and results are interesting. However, there are some issues that require improvement or clarification before the work is accepted.

1.      The antimicrobial activity values ​​of compounds 3a-6c from Table 1 (T. mentagrophytes) and Table 2 (MRSA) are identical. Was there a mistake in the presentation of the data? If so, the discussion and conclusions must be corrected.

2.      Is it possible to add to tables 1 and 2 values of antimicrobial activity ​​obtained for lophanic acid? This would complete the structure-activity analysis of its derivatives.

3.      Reference 15 is missing in the manuscript.

4.      The chemical names of the obtained compounds are missing.

5.      There are many peaks in HRMS spectrum of compound 6b. Can you show the purity of this compound by HPLC?

6.      In the 13C NMR data for compound 6g, data of the two aromatic carbon are missing.

7.      Line 68: I suggest replacing “14” to fourteen (subsequent numbers in this sentence merge together).

8.      Inconsistency in the spelling of strains of bacteria and fungi - sometimes italic, sometimes normal font.

Reviewer 2 Report

A very interesting publication, however, I have a few comments.

From what I can see in compounds 5  there is no ester but an ether bond (line 71).

Why did the authors limit themselves to testing the antimicrobial activity in only one concentration of the compounds? Why were the MIC and MBC values not determined? This may give information about the bactericidal or bacteriostatic effects of the compounds.

Author Response

Reviewers

Comments Editor/Reviewers

Response

Reviewer 2

1. From what I can see in compounds 5  there is no ester but an ether bond (line 71).

Thanks for the suggestions. We have corrected that accordingly in the article.

2. Why did the authors limit themselves to testing the antimicrobial activity in only one concentration of the compounds? Why were the MIC and MBC values not determined? This may give information about the bactericidal or bacteriostatic effects of the compounds.

Thanks for your suggestions.

The initial antimicrobial screening concentration was set at 100 μg/mL. For the very active samples, we did the further microbroth dilution assay to get MIC or MBC. However, the active compounds (against MRSA), such as 3b, 3d, 5c and 5h with inhibition rate above 90%,  the inhibition rates were <60% at the first dilutions (50 μg/mL). It indicates that the MICs and MBCs are ≥ 100 μg/mL. Thus, we did not add the MIC and MBC in the table.

Others

All changes in the manuscript  were highlighted in yellow.

Round 2

Reviewer 1 Report

The authors have included the suggested corrections in the manuscript. The sentence from lines 101-103 still needs to be improved, because the compounds 3e and 5h after the change in table 1 should not be included in it.

Author Response

Comments:

The authors have included the suggested corrections in the manuscript. The sentence from lines 101-103 still needs to be improved, because the compounds 3e and 5h after the change in table 1 should not be included in it.

Responses:

We thank the reviewer's valuable suggestions. 

In line 101-103: The sentence "We further observed that the introduction of the ester groups at C-20 could improve the bioactivity potency of the lophanic acid derivatives (e.g., 3a-3d, 5a and 5b)." has been changed into "Based on the antimicrobial  activities of the lophanic acid derivatives (e.g., 3a, 3d, 5a and 5b), the structural modification at C-20 could be performed to improve the bioactivity."

Reviewer 2 Report

The authors' answers are sufficient.

Author Response

We thank the reviewer's efforts for the improvement of our manuscript.